# Toward Robust Real-World Audio Deepfake Detection: Closing the Explainability Gap

## Abstract

The rapid proliferation of AI-manipulated or generated audio deepfakes poses serious challenges to media integrity and election security. Current AI-driven detection solutions lack explainability and underperform in real-world settings. In this paper, we introduce novel explainability methods for state-of-the-art transformer-based audio deepfake detectors and open-source a novel benchmark for real-world generalizability. By narrowing the explainability gap between transformer-based audio deepfake detectors and traditional methods, our results not only build trust with human experts, but also pave the way for unlocking the potential of citizen intelligence to overcome the scalability issue in audio deepfake detection.

## 1 Introduction

The rapid proliferation of AI-generated audio deepfakes poses a growing threat to media integrity, personal security, and democratic processes, with critical implications for misinformation, fraud, and election security. Although state-of-the-art detection solutions have demonstrated promising results on benchmark datasets, they often fall short in real-world scenarios due to poor generalization and a lack of explainability. Current approaches, such as those used in ASVspoof and related competitions, focus heavily on detection performance within constrained environments, but their efficacy diminishes significantly when encountering diverse and unseen samples.

In this work, we highlight the limitations of existing deepfake detection methods and introduce an attention roll-out mechanism that addresses these shortcomings by providing improved explainability for transformer-based audio classifiers. Recent deepfake audio attacks, such as those used to discredit Marti Bartes in Mexico or to influence U.S. elections by impersonating President Joe Biden, emphasize the necessity for solutions that not only detect these manipulations but also offer transparent and interpretable explanations (Goodman, 2024; Staff, 2024). These would, in turn, foster trust with both experts and the general public.

To bridge the gap between controlled benchmark results and real-world applicability, we deliver a novel benchmark that evaluates the generalization capabilities of deepfake audio classifiers by training on the ASVspoof 5 dataset and testing on the FakeAVCeleb dataset. This benchmark provides a more realistic evaluation of model robustness, simulating conditions where the data distributions of training and testing are substantially different. Additionally, we compare and contrast various explainability methods, offering a conceptual contribution that defines key requirements for explainability in deepfake audio detection. Our findings not only reveal the strengths and weaknesses of each approach but also lay the groundwork for future research.

Through this comprehensive evaluation, we outline several open challenges that need to be addressed to improve generalization and explainability, thereby enhancing trust in deepfake detection systems. By establishing these benchmarks and defining conceptual requirements, we hope to catalyze future developments in this critical research area to effectively safeguard against the evolving threat of audio deepfakes.

We list our novel contributions as follows:

- A conceptual explainability framework for deepfake audio detection (Section 4).
- Empirical evaluations of novel explainability methods for audio transformers (Section 5).
- A generalizability benchmark for deepfake audio classifiers (Section 6).

## 2 RELATED WORK

**Traditional Machine Learning Approaches**  The early days of deepfake audio detection were dominated by traditional classifiers, such as support vector machines (SVM) and Gaussian mixture models (GMM), and traditional signal processing features. These works generally use subsets of the various hand-crafted features described above, and they generally perform well with academic datasets, in which the distributions of features between real and deepfake audio are relatively easily separable. Observing that these models typically do not generalize well when presented with deepfake audio from unseen distributions, Zhang et al. (2021) propose using an SVM to learn a tight boundary around the features of real audio by only training on real audio (Zhang et al., 2021). This method outperforms most other methods on the ASVspoof 2019 dataset, but it is unlikely to perform as well when the true audio varies more widely (e.g., non-English speech, speakers with accents, non-adult speakers, etc.).

Recent work also suggests that using an ensemble of gradient-boosting decision trees (GBDT) may be more robust to unseen data as well as boast faster inference times than both SVM and GMM (Bird & Lotfi, 2023). In a 2023 work by Bird & Lotfi (2023), the authors demonstrate the power of the GBDT, reporting accuracies of 99.3% on the DeepVoice dataset and inference times of 0.004 seconds for 1 second of input speech (Bird & Lotfi, 2023). In their work *Real-Time Detection of AI-Generated Speech for Deepfake Voice Conversion*, they also explore features importances and the statistical characterizations of real and deepfake audio. Another recent work by Togootogtokh & Klasen (2024) employs a GBDT for deepfake audio classification task with a custom dataset comprised of true samples from the LJ Speech Dataset and deepfake samples generated with various HuggingFace TTS models (Togootogtokh & Klasen, 2024).

**Self-Supervised Embedding-Based Classifiers**  A few recent works use self-supervised embedding features as the basis of their classification algorithms, most of which use Wav2Vec features (Shi & Yamagishi, 2021). Tak et al. (2022) fine-tune a transformer model with a Wav2Vec front-end and report the lowest equal error rates for both the ASVspoof 2021 Logical Access and Deepfake databases. Xie et al. (2021) use Wav2Vec features as input to a Siamese neural network that they train to distinguish whether the speech samples in a pair belong to the same category. This work reported state-of-the-art results on the ASVspoof 2019 dataset (Chen et al., 2022). Some other recent works use HuBERT features, and Wang & Yamagishi (2022) compare Wav2Vec-, XLS-R-, and HuBERT-based features (Yi et al., 2023). Most recently, Le et al. (2024) combine AST features with a GBDT ensemble to detect deepfake audio, which they plan to use for a continuous learning approach.

**Explainability for Audio**  The literature for audio explainability is limited (Akman & Schuller, 2024). Though general explainability methods such as LIME and SHAP can also be used for models trained on audio data, very few examples of attempts for audio explainability exist in the literature (Ribeiro et al., 2016; Lundberg & Lee, 2017). Of those exceptions, Yanchenko et al. (2021) measure the similarity of deep features to hand-crafted features and attempt explain deep convolutional features as they relate to traditional signal processing ones. Most recently, Becker et al. (2024) introduced AudioMNIST, a novel audio dataset consisting of 30 000 audio samples of spoken digits, and proposed using Layer-wise Relevance Propagation (LRP) to explain neural network classifications. They also investigate using a combination of visual and aural explanations, and find aural explanations promising if well-designed.

For additional related work on in-painted deepfake audio and convolutional neural network (CNN) based approaches to deepfake audio detection, refer to Appendix A

## 3 BACKGROUND

### 3.1 TRADITIONAL METHODS

In the previous section, we reviewed a variety of traditional methods for deepfake audio detection. Here, we will focus on features used in traditional methods and define the gradient boosting decision tree method introduced in the previous section.

**Signal Processing Features**  Traditional methods typically rely on "hand-crafted" features, which are not learned by neural networks but calculated with standard signal processing methods.

Mel-Frequency Cepstral Coefficients (MFCCs) are a set of 10-20 features that capture the timbre of audio samples using the perceptual Mel scale, reflecting how humans perceive pitch. Computing MFCCs involves applying a pre-emphasis filter, dividing the signal into 20 ms frames, applying a Hann window, and performing a Fast Fourier Transform (FFT). The resulting power spectrum is passed through a Mel-scale filter bank, then logarithmically transformed, and a Discrete Cosine Transform (DCT) is applied to generate MFCCs, providing a compact representation of spectral properties. Other spectral features include spectral centroid (indicating "brightness" by representing the center of spectral mass), bandwidth (spread around the centroid), and roll-off (frequency below which 85% of spectral energy is contained). These features are useful for tasks like music genre classification and speech analysis.

Chroma features capture harmonic content by mapping the spectrum to twelve pitch classes (C, C#, D, etc.) and are invariant to timbre changes, making them ideal for analyzing musical elements. Zero Crossing Rate (ZCR) measures the rate of sign changes in an audio signal, aiding in pitch and speech detection, while Root Mean Square (RMS) captures energy or loudness, offering a measure of signal strength.

**Gradient Boosting Decision Trees**  Gradient boosting decision trees (GBDT) combine three core concepts in traditional machine learning: ensemble learning, boosting, and gradient descent. As an *ensemble* method, GBDT combine the predictions of several weak learners–typically decision trees–to produce a stronger overall prediction. The *boosting* aspect of GBDT means that the model is constructed sequentially, such that at each iteration a new weak learner is added to correct for the previous learners' mistakes. Finally, the *gradient* aspect of GBDT reflects the optimization technique used to find the best fit for each new weaker learner added to the ensemble.

At initialization, the GBDT is an ensemble containing a single weak learner that makes a prediction. For binary classification, the model is initialized with a constant value $\hat{F}_0(x)$, the log-odds of the positive class. At each iteration, until the maximum number of weak learners permitted in the ensemble is reached, the pseudo-residuals are calculated, which are the negative gradients of the loss function:

$$r_{im} = -\left[\frac{\partial L(y_i, \hat{F}_{m-1}(x_i))}{\partial \hat{F}_{m-1}(x_i)}\right] = y_i - \hat{p}_{m-1}(x_i), \tag{1}$$

where $r_{im}$ is the residual for the $i$-th instance at iteration $m$, $L(y_i, \hat{F}_{m-1}(x_i))$ is the loss function to be minimized, $\hat{p}_{m-1}(x_i)$ is the predicted probability of the positive class for the $i$-th instance at iteration $m-1$, and $y_i$ is the true label. A new weak learner, $h_m(x)$, is then fitted to this residual and its impact scaled to avoid overfitting, such that the model at iteration $m$ is given by:

$$\hat{F}_m(x) = \hat{F}_{m-1}(x) + \nu \cdot h_m(x), \tag{2}$$

where $\nu$ is the learning rate. At the end of the learning process, the predicted probability of a data sample's membership in the positive class is given by:

$$\hat{p}(x) = \frac{1}{1 + \exp(-\hat{F}_m(x))}. \tag{3}$$

The final classification is given by thresholding the prediction probability $p$.

## 3.2 TRANSFORMERS

Since the publication of *Attention is All You Need* (Vaswani et al., 2017), transformer models have become increasingly widespread for a wide variety of tasks, though most notably text generation. As suggested by the title of that 2017 paper, *attention* is the core of the transformer architecture. Given a sequence of input embeddings $E$, a transformer model encodes tokens using a *self-attention* mechanism, which allows the model to focus on different parts of the input sequence when encoding a particular token (Vaswani et al., 2017). A single self-attention operation (or head) is defined by:

$$\text{Attention}(Q, K, V) = \text{softmax}\left(\frac{QK^T}{\sqrt{d_k}}\right)V, \tag{4}$$

where $\mathbf{X}$ is the matrix of input embeddings, $Q = \mathbf{X}W_Q$ is called the query matrix, $K = \mathbf{X}W_K$ is called the key matrix, $V = \mathbf{X}W_V$ is called the value matrix, $W_Q, W_K, W_V$ are learned weight matrices, and $d_k$ is the dimensionality of the keys (Vaswani et al., 2017). In order to facilitate learning multiple different features, multiple self-attention heads are used. Multi-attention is then defined by:

$$\text{MultiHead}(Q, K, V) = \text{Concat}(\text{head}_1, \ldots, \text{head}_h)W_O, \tag{5}$$

where $\text{head}_i = \text{Attention}(QW_{Q_i}, KW_{K_i}, VW_{V_i})$ and $W_O$ is the output projection matrix. The output of the multi-head attention is passed into a Feed-Forward Neural Network (FFN) defined by:

$$\text{FFN}(x) = \text{ReLU}(xW_1 + b_1)W_2 + b_2, \tag{6}$$

where $W_1$ and $W_2$ are learned weight matrices, $b_1$ and $b_2$ are biases, and ReLU is the activation function. Each multi-head attention or FFN sub-layer is followed by layer normalization defined by:

$$\text{Layer Output} = \text{LayerNorm}(x + \text{Sub-layer}(x)). \tag{7}$$

The final transformer architecture stacks multiple of these multi-head attention and FFN layers to capture increasingly complex patterns.

Since the transformer network does not maintain input order, an additional positional embedding token is appended to each patch to allow the model to maintain the spatial structure of the input spectrogram (Gong et al., 2021). We also often prepend a `[CLS]` token to the input. After passing through all transformer layers, the final `[CLS]` token aggregates information from the entire input sequence for the final prediction.

The aforedescribed operations are universal to the transformer architecture, but methods of creating the input sequence vary widely. The transformer architecture relies on the creation of *tokens* from raw input data and the learned *attention* between those tokens. For natural language tasks, tokens typically represent individual words. For image tasks, tokens are typically pixel patches. For audio tasks, there are a variety of approaches. Here, we introduce two popular mechanisms for generating input embeddings from audio data.

**Self-Supervised Audio Features**  One of the most popular feature generators is Wav2Vec 2.0, produced and published by Meta in 2019 (Schneider et al., 2019). Wav2Vec uses a 7-layer CNN generate latent feature encodings, which are then put into a quantization module to make the final tokens which will be fed to the transformer (Schneider et al., 2019). As speech is continuous, Wav2Vec strives to automatically infer discrete speech units with the quantization module, such that tokens can be formulated as they would be in natural language, representing complete but discrete data units (Schneider et al., 2019).

In contrast, the Audio Spectrogram Transformer (AST), which was the first to move away from traditional convolutional neural network approaches, simplifies audio token generation (Gong et al., 2021).

As seen in Figure 1, the AST first transforms an input audio wave of length $t$ seconds into a sequence of 128-dimensional Mel features. The resultant $128 \times 10t$ spectrogram is then used as input to the AST. The spectrogram is then split into a sequence of $N$ $16 \times 16$ patches, with overlap in both time and frequency dimensions. Each $16 \times 16$ patch is flattened into a single-dimensional patch of size 768 with a linear layer.

Once the audio is formatted in this way, the AST feeds the input sequence to a Vision Transformer (ViT), an image transformer model trained on ImageNet (Gong et al., 2021). This approach essentially translates the audio signal into an image and then uses a transformer pre-trained on image data to make classifications.

Notably, both the AST and Wav2Vec models are pretrained only with real audio data, only by finetuning additional layers are they suitable for classifying deepfake audio.

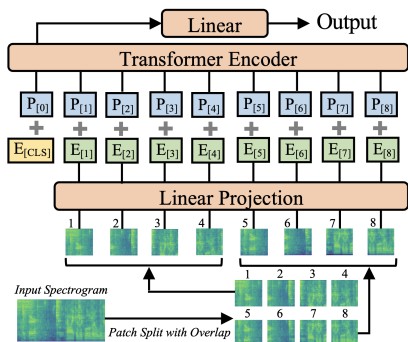

Figure 1: Diagram of the audio spectrogram transformer architecture introduced by Gong et al. (2021)

## 3.3 EXPLAINABILITY AND INTERPRETABILITY

Though *explainability* and *interpretability* are often used interchangeably, we distinguish between them. Interpretability emphasizes transparency and clarity, focusing on making the internal mechanisms of a model comprehensible, such as understanding the coefficients of a linear regression model or the structure of a decision tree (Rudin, 2019). In contrast, explainability goes beyond understanding the internal, mathematical mechanism and should provide explanations of a model's predictions in a human-understandable way, even if the model itself is a complex "black box" (Rudin, 2019).

Explainability oftens involves using post-hoc methods, LIME (Local Interpretable Model-agnostic Explanations) or SHAP (SHapley Additive exPlanations), to approximate and elucidate the model's decisions without revealing the exact inner workings (Ribeiro et al., 2016; Lundberg & Lee, 2017). Importantly, the interpretability of a model can contribute to its explainability, but a model's being interpretable does not necessarily imply that it is explainable.

**Explainability for GBDT** A key advantage of traditional methods is that they are interpretable. With a gradient boosting decision trees, or any other tree-based ensemble method, feature importances can be calculated. As described in Algorithm 1, feature importances are calculated by measuring the model's change in performance after each feature is permuted (Pedregosa et al., 2011). To stabilize results, we permute each feature multiple times and use the mean and standard deviation of each feature's importance.

However, the importance of a given feature may be obscured by the permutation feature importance algorithm if multiple features are multicollinear, as is the case for the audio signal features described in the previous section. Intuitively, permuting one feature will have little impact on the model's performance if the same, or very similar, information is available to the model through another non-permuted feature. To combat this issue, we perform hierarchical clustering on the Spearman rank-order correlations between features and keep a single feature from each cluster. This way, when a feature is permuted, there should be no other non-permuted feature containing duplicate information.

**Explainability for Transformer Models** A challenge of working with transformer methods is the lack of interpretability. Though they boast much better performance than traditional methods, again see Appendix D, their output is of a "black-box" nature. In order to facilitate citizen intelligence, detection methods must deliver human interpretable explanations that are sample-specific, such that the explanation is not invariant to the input sample; time-specific, such that the explanation includes specific timestamps that localize the distinguishing features; and feature-specific, such that specific aspects like unusual noise or errant formants can be identified as unnatural.

## 3.4 MODEL PERFORMANCE FOR DEEPFAKE AUDIO CLASSIFICATION

As mentioned in the previous section, the impressive performance of Wav2Vec-based transformers has already been demonstrated by Tak et al. (2022). Le et al. (2024) recently presented similar results using AST features with a GBDT. To deliver on the goal of explainable results that maintain the performance of transformer-based methods, we validate the performance of the finetuned AST model, the finetuned Wav2Vec model, and the traditional feature-based GBDT on the ASVspoof5 and FakeAVCeleb datasets.

These results, as shown in Appendix D, demonstrate the superior performance of the Wav2Vec and AST models as well as report the GBDT baseline. Though the power of a finetuned Wav2Vec transformer has already been established, the results reported in Appendix D are state-of-the-art for the FakeAVCeleb dataset. For the sake of robustness, we also report results on the ASVspoof 5 dataset when the data has been compressed and rerecorded to measure the effect that these common data augmentations have on model performance.

## 4 METHODS

In this section, we introduce our proposed methods for audio explainability, with which we will experiment in the following section. We appropriate methods for vision and natural language explainability and translate them to the audio domain.

### 4.1 OCCLUSION

Occlusion is a technique used for vision model explainability, particularly with deep learning models that might otherwise be considered "black boxes". The core idea is to iteratively occlude, or block from view, parts of the input data, measure how the model's prediction changes, and, ideally, identify which parts of the input data are most important.

Consider some input $X = [x_1, x_2, \ldots, x_n]$, where $X$ represents the original input and each $x_i$ represents a subsection (perhaps a pixel, patch, or token) in $X$, and some model $f$. First, we generate a baseline prediction $\hat{y} = f(X)$, which will serve as the point of comparison. Then, we iteratively mask each $x_i$ from the input, such that:

$$\mathbf{X}^{(i)}_{\text{occluded}} = \mathbf{X} \odot \mathbf{M}_i, \tag{8}$$

where $\mathbf{M}_i$ is a mask that occludes the $i$-th subregion of the input and $\odot$ represents element-wise multiplication. After this operation, the occluded region will be replaced with some specified value (e.g., 0, 1, or a mean of the feature across all samples). For each occluded input, the model makes a new prediction given by:

$$\hat{y}^{(i)} = f(\mathbf{X}^{(i)}_{\text{occluded}}). \tag{9}$$

The intuition is that if a change in the model's prediction is observed when a region is occluded, that region is likely important. After occluding different regions and observing changes in the model's behavior, the results can be visualized in a heatmap. This method has been used for understanding importance in vision data, but we introduce it here for audio data. We treat the Mel-spectrogram representation of each audio sample as an image and occlude sections of the spectrogram to determine which parts of each audio sample are most important for the transformer's classification. This method delivers on all three aspects of sufficient explainability defined in 3.3.

### 4.2 ATTENTION VISUALIZATION

We also appropriate an explainability method introduced for use with natural language. As previously discussed, transformer models rely on a self-attention mechanism to understand the relationships between different parts of the input sequence. The attention mechanism assigns a weight to each token, which reflects the importance of each token in relation to every other token.

Recall that attention is defined by:

$$\text{Attention}(Q, K, V) = \text{softmax}\left(\frac{QK^T}{\sqrt{d_k}}\right)V, \tag{10}$$

where $\mathbf{X}$ is the matrix of input embeddings, $Q = \mathbf{X}W_Q$ is called the query matrix, $K = \mathbf{X}W_K$ is called the key matrix, $V = \mathbf{X}W_V$ is called the value matrix, $W_Q, W_K, W_V$ are learned weight matrices, and $d_k$ is the dimensionality of the keys (Vaswani et al., 2017). After applying a softmax on the unnormalized attention weights, we are left with a normalized $d_k \times d_k$ matrix of weights that can be visualized as a heatmap. In such a visualization, the $x - axis$ represents position in the input sequence (or token ID), the $y - axis$ represents the tokens for which attention is computed, and the color intensity at some $(i, j)$ represents the attention weight for token $i$ on token $j$, where greater attention is read to reflect higher importance.

A limitation of attention visualization is that it is done per layer per head, which can make it difficult to observe the overall model's attention. To combat this, Abnar & Zuidema (2020) proposed "attention rollout" to trace the distribution of attention across multiple or all of the model's layers. This method gives us a more complete view the model's distribution of attention.

Consider an $L$-layer transformer with attention matrices $W_l$ for $l \in \{1, 2, \ldots, L\}$, where each $W_l$ represents the attention between different tokens at that layer. We compute a cumulative attention

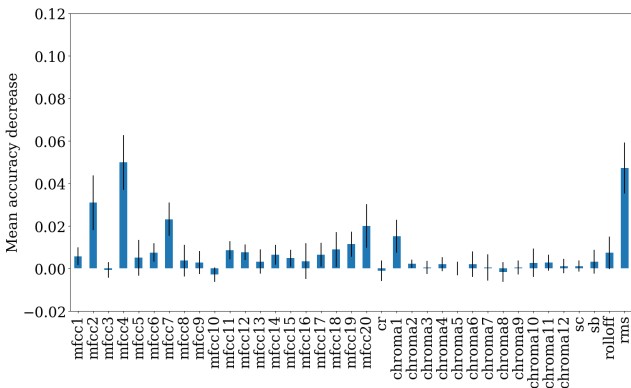

Figure 2: GBDT feature importances as measured by mean accuracy decrease with standard deviations for 6.0-second classifier.

matrix by multiplying the attention matrices across the layers, such that:

$$W^{\text{rollout}} = W^{(1)} W^{(2)} \dots W^{(L)}. \tag{11}$$

Once we have computed the cumulative attention matrix $W^{\text{rollout}}$, it can be visualized similarly to the single attention weights. We also extract the attention weights specific to the `[CLS]` token to understand what parts of the input sequence were most relevant for the final classification. Attention visualization and roll-out were introduced for use with natural language tokens. In this paper, we adapt this method for use with audio tokens.

## 5 EXPERIMENTS

We experiment with three models: an ensemble of gradient boosting decision trees (GBDT), an AST-based transformer, and a Wav2Vec-based transformer. An extended discussion of the ASVspoof 5 and FakeAVCeleb datasets can be found in Appendix B, while a report of hyperparameters for each of our three models can be found in Appendix C.

### 5.1 EXPLAINABILITY FOR GBDT

A well-noted advantage of ensembles of decision trees, as the GBDT is, is the ability to calculate feature importances. In their recent deepfake audio classification with GBDT work, Bird et al. report feature importances and draw meaning from them Bird & Lotfi (2023). Here, we mimic their approach to explain the behavior of the GBDT and to isolate some aspect of the audio sample as a signature of its deepfake classification.

We compare the GBDT feature importances, calculated with the permutation importance algorithm, for the models trained with 1.0, 3.0, and 6.0-second audio samples (Pedregosa et al., 2011). The results for 6.0-second classifier are shown in Figure 2, while supplementary results for the 1.0- and 3.0-second classifiers can be found in subsection E.1. As seen in Figure 2, the second MFCC, fourth MFCC, and RMS features are the most influential in the GBDT's decision-making. Recall that the RMS feature is most closely associated with the loudness of an audio sample. We find the high importance of the RMS troubling as our intuition suggests that loudness should not inherently be a characteristic of deepfake audio.

We retrain the GBDT with only the three most important features, the second MFCC feature, the fourth MFCC feature, and the RMS, and evaluate the model's performance when given only these three features. We observe some performance degradation; when asked to classify 6.0-second audio samples, the model is only able to achieve 70.0% precision, recall, and accuracy (compared to 89.0% precision, recall, and accuracy when trained with all features).

As the vast majority of features are estimated to have a less than 2% impact on overall accuracy, we also calculate feature correlations. As shown in Figure 3, many of the features are correlated.

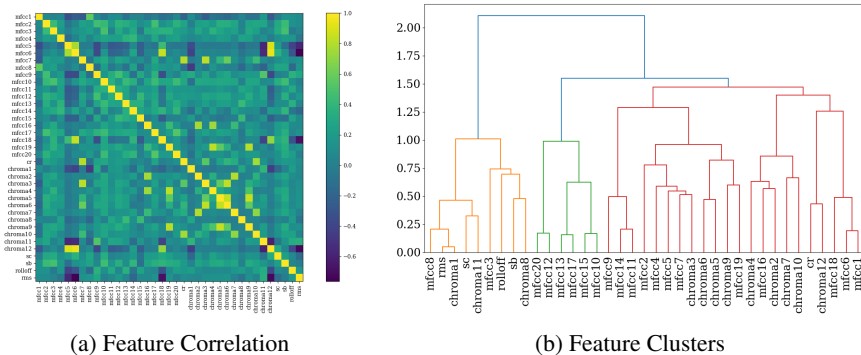

(a) Feature Correlation           (b) Feature Clusters

Figure 3: GBDT feature correlations and clusters for 6.0-second classifier. In these figures, `sc` refers to the spectral centroid, `sb` refers to the spectral bandwidth, `cr` refers to the ZCR, mfcc$i$ refers to the $i$-th MFCC feature, and chroma$i$ refers to the $i$-th chroma feature.

We perform a hierarchical clustering of the features using Ward's linkage, and observe that there are only a few clusters of features. Perhaps interestingly, there is no clustering of the features in which our three most important features, MFCC2, MFCC4, and RMS, are all in different clusters.

We select the most important feature from each cluster and retrain the GBDT. Retrained with RMS, MFCC 20, and MFCC 4, the GBDT achieves precision of 64.3%, recall of 64.0%, and accuracy of 63.8%. As performance was better when using the three most important features, compared to using important features with more spread, it does seem that the second MFCC feature is actually critical to the model's decision-making. Though it is not feasible to attribute a single frequency to a single MFCC, as an MFCC is a compact representation of a spectral shape across the Mel-scale filterbank, the second MFCC captures low-frequency details, such as overall spectral slope and formant information. Each formant corresponds to a resonance in the vocal tract, and it is intuitive that deepfake audio would have anomalous resonance. While this explanation points to a potential source of inherent distinction between real and deepfake audio, it only provides an explanation as to what the model is attentive to in general rather than sample-level specificity. While the GBDT is interpretable, we find that it is not sufficiently explainable to be useful to a non-technical audience.

## 5.2 EXPLAINABILITY FOR AUDIO TRANSFORMERS

**Occlusion**  As the AST model converts the raw audio signals into a spectrogram input, it is well-suited to visualization. We perform occlusion with box size $(200, 50)$ and stride size $(100, 25)$. Importance is measured by the magnitude of change in the predicted probability of the sample being in the positive class when a section is occluded.

As shown in Figure 4, the importance is greatest for the padded regions–regions that theoretically contain no predictive information. The audio samples shown in Figure 4 are of length 6.0-seconds, but we observe this phenomenon when calculating importance by occlusion for all sample lengths. For each sample length, the most important regions (as measured by the occlusion method) are always placed at the beginning of the padded region. This result is obviously unhelpful in explaining the model's decision-making, but it is suggestive of a phenomenon posited by Wu et al. (2021), in which transformer models store global information at locations in the feature space which are consistent, such that the weight information there is always propagated (Wu et al., 2021).

**Attention Visualization**  Another approach to explaining transformer models is visualizing the distribution of the attention weights over the input data. For each layer, as described in Section 3, there is an attention matrix that represents the amount of attention between each pair of tokens. This method has been employed for image and text data, but not, as of yet, to audio data (Dosovitskiy et al., 2021).

We visualize the each layer's attention matrix for the same bonafide and spoofed samples shown in Figure 4. The resultant visualizations can be found in subsection E.2. Similar to results recorded on Vision Tranformer (ViT) by Dosovitskiy et al. (2021), we observe that the attention at early

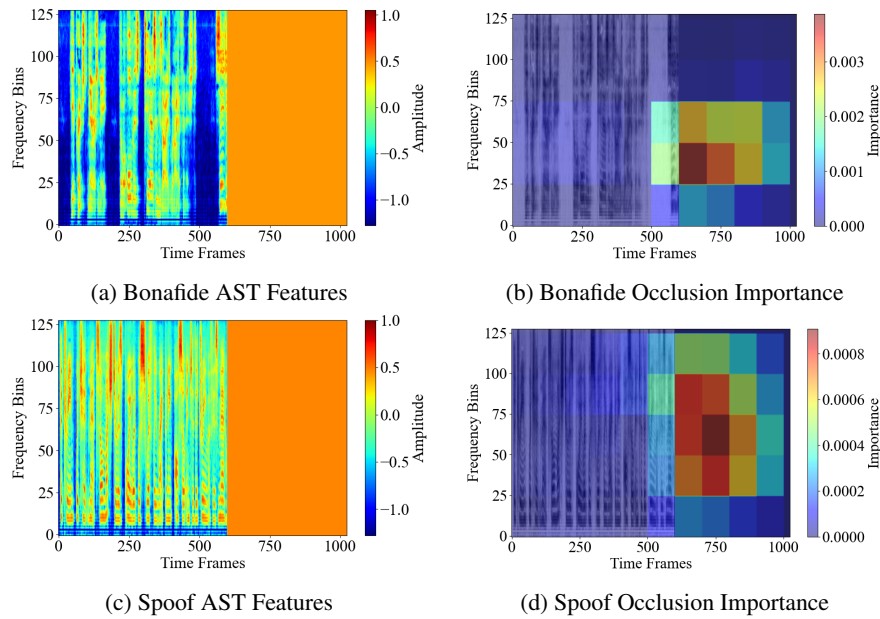

(a) Bonafide AST Features

(b) Bonafide Occlusion Importance

(c) Spoof AST Features

(d) Spoof Occlusion Importance

Figure 4: Importance measured by occlusion for 6.0-second audio samples.

layers is quite local with a relatively small receptive field while the attention at later layers is widely distributed (Dosovitskiy et al., 2021).

To better understand the distribution of attention across the entire model, we compute the attention roll-out, which allows us to observe the overall attention flow on each of the input tokens by recursively multiplying the weight matrices of all the layers (Abnar & Zuidema, 2020).

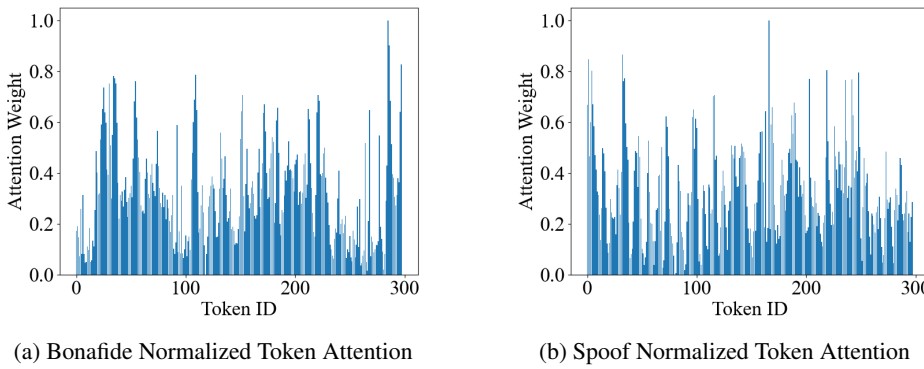

(a) Bonafide Normalized Token Attention

(b) Spoof Normalized Token Attention

Figure 5: Distribution of attention for 6.0-second audio samples.

By normalizing the attention for the `[CLS]` classification token, we are able to visualize which input tokens are most important for the model's overall classification, as seen in Figure 5. As each Wav2Vec token represents about 20 milliseconds of audio signal, we can pinpoint specific frames that were instrumental in the classification and inspect them more closely. Figure 5 allows us to identify very short audio frames that are most influential in the model's prediction, and we observe that influential tokens typically appear in groups.

## 6 GENERALIZABILITY BENCHMARK

Finally, we introduce a novel benchmark to evaluate the generalization capabilities of deepfake audio classifiers to unseen data, which is critical for deploying reliable deepfake detection systems.

This benchmark measures the robustness and transferability of models across different datasets with varying characteristics, simulating real-world scenarios where deepfake classifiers may encounter out-of-domain samples.

We train each of our models with the ASVspoof 5 dataset and evaluate each model's ability to classify samples from the FakeAVCeleb dataset. As a reminder, the evaluation performance of each model on ASVspoof 5 and FakeAVCeleb independently can be found in Appendix D. We compare performance between our baseline classifier, the GBDT, and the two transformer methods. We use 3 000 evaluation samples from FakeAVCeleb, balancing the classes to an equal number of both bonafide and spoof audio samples.

Table 1: Performance comparison of ASVspoof-trained models on FakeAVCeleb data.

| Model | Class | Precision | Recall | F1 |
|---|---|---|---|---|
| GBDT | bonafide | 0.50 | 0.58 | 0.54 |
| | spoof | 0.51 | 0.51 | 0.51 |
| AST | bonafide | 0.85 | 0.84 | **0.85** |
| | spoof | 0.85 | 0.86 | **0.85** |
| Wav2Vec | bonafide | 0.73 | **0.98** | 0.84 |
| | spoof | **0.97** | 0.63 | 0.77 |

When using the GBDT classifier trained on 6.0-second samples of the original ASVspoof dataset, we observe that an overall accuracy of 51%, which indicates that the GBDT does essentially no better than random guessing between the two classes. Table 1 reports the precision, recall, and F1 scores for both bonafide and spoof audio for all three models. The transformer methods perform much better: the AST and Wav2Vec models achieve an overall accuracy of 85% and 81%, respectively, on the FakeAVCeleb evaluation data. Wav2Vec is by far the most popular feature encoder in the literature, likely due to its generally superior performance, but here the AST-based transformer offers much better balanced performance on the out-of-distribution data.

## 7 DISCUSSION AND CONCLUSION

In this paper, we address the critical issue of audio deepfake detection by introducing a novel benchmark that evaluates the generalization capabilities of state-of-the-art transformer-based models. Our experiments, conducted using the ASVspoof 5 and FakeAVCeleb datasets, demonstrate that current detection solutions often struggle with generalizability and lack sufficient explainability, especially in real-world scenarios. By incorporating explainability methods such as attention roll-out and occlusion, we highlight the strengths and limitations of these approaches, providing a clearer understanding of model decisions.

The attention roll-out mechanism, in particular, shows promise in visualizing transformer-based models' attention across multiple layers, enabling a more transparent analysis of how decisions are made. Our results indicate that while transformer models like Wav2Vec and AST outperform traditional models on unseen data, there remains a significant gap in their ability to provide human-understandable explanations. This finding underscores the need for more research in this area, especially to improve interpretability for non-technical users and domain experts.

Looking ahead, future work should focus on refining explainability methods for transformer-based models and developing new benchmarks that further challenge the robustness and interpretability of deepfake detectors. Additionally, understanding and mitigating the effect of data augmentations such as compression and re-recording will be essential for creating more resilient models. A limitation of this study is the reliance on only two datasets, which may not capture the full range of manipulations seen in real-world deepfake audio. Further limitations are addressed in Appendix F. Moreover, the proposed explainability methods are still in their infancy and may not yet offer insights that are as intuitive to non-technical users. By addressing these challenges, we can move closer to building reliable, trustworthy audio deepfake detection systems that are ready for real-world deployment.

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

## A RELATED WORK (CONTINUED)

### A.1 CNN-BASED CLASSIFIERS

The most popular architecture for deepfake audio detection is by far the convolutional neural network (CNN) based classifier Dixit et al. (2023); Wu et al. (2020); Lavrentyeva et al. (2019); Zeinali et al. (2019); Dinkel et al. (2017). CNN classifiers have been historically favored for their ability to combine spatial and temporal information through convolution. The most successful of these is called *Light Convolutional Neural Network* (LCNN) and was proposed by Wu et al. in 2020 Wu et al. (2020). LCNN consists of convolutional and max-pooling layers with Max-Feature-Max (MFM) activation. Wu et al. choose the MFM activation function instead of the arguably standard ReLU function because they observe that the MFM learns more compact features than the sparse high-dimensional ones learned with ReLU Wu et al. (2020). This distinction is what makes their CNN "light". The LCNN has been incredibly successful in recent ASVspoof and ADD competitions; it was the best system at ASVspoof 2017 and continued to be the best system in one of the sub-tasks of ASVspoof 2019. Another successful CNN architecture, proposed by Dinkel et al., uses raw waveforms as input to a convolutional long short-term neural network. Their model combines time-convolving layers with frequency-convolving layers to "reduce time and spectral variations" with long-term temporal memory layers to capture longer-term temporal relationships Dixit et al. (2023); Dinkel et al. (2017). A variety of other CNN architectures have been proposed, but all of them generalize poorly to unseen attacks Dixit et al. (2023).

### A.2 DEEPFAKE AUDIO IN-PAINTING

Another relevant research area is that of deepfake audio in-painting detection. In-painting refers to the practice of mixing real and fake audio such that only small portions of an audio sample are actually manipulated. Detecting deepfake audio in-painting requires not only identifying that a sample contains some corrupted audio but also identifying the timestamps at which the corruption begins and ends. Xie et al. (2024) propose a framework called EAT, which incorporates a ResNet, a two-layer transformer encoder, a single-layer bidirectional Long Short-Term Memory network (LSTM), and a final classification layer.

The EAT framework achieves an F1 score of 98% when classifying segments at 20 millisecond resolution Xie et al. (2024). However, Xie et al. only evaluate their method on a custom dataset, do not attempt to evaluate their architecture's performance on any of the standard deepfake audio benchmarks, and focus primarily on deepfake environmental sounds and background audio Xie et al. (2024). A slightly older but broader work, from 2022, that investigates in-painted deepfake audio is that of Cai et al. (2022) in *Waveform Boundary Detection for Partially Spoofed Audio*. Cai et al. (2022) use a combination of Wav2Vec and MFCC features as their input to a series of single-dimensional CNNs, a transformer encoder, a bidirectional long short-term memory network (BiLSTM), and a final linear layer for classification (Cai et al., 2022). With their method, they achieve the best performance in the locating manipulated clips task of the ADD 2022 challenge.

## B DATA SOURCES

We employ a variety of publicly available datasets to conduct our experiments. We also craft custom variations in order to better mimic real-world use cases and challenges.

The majority of our experiments are conducted with the ASVspoof 5 dataset. The ASVspoof5 dataset is a state-of-the-art dataset containing eighteen different varieties of deepfake audio as well as true speech samples (Wang et al., 2024). Each sample is labelled with a classification as "bonafide" or "spoof". If the sample is spoofed, the attack method that was used to generate the sample is specified. The dataset, which was released in June 2024, contains $182\,357$ train samples and $142\,134$ test samples. Each deepfake (or spoofed) audio sample is generated with a novel VC or TTS method, which were trained on two English-language datasets. The final deepfake samples are made using the English-subset of the Multilingual LibriSpeech (MLS) dataset (Wang et al., 2024). The 18 different attack types included in this dataset make it the most attack diverse of the datasets we consider in this study. We take this dataset to represent the state-of-the-art in deepfake audio generation and, in subsequent sections, refer to it simply as the ASVspoof dataset.

The audio samples provided by ASVspoof are clean. The ASVspoof bonafide samples were created in recording studios with high quality microphones; The ASVspoof fake samples were generated with TTS algorithms that add no additional noise. When deepfake audio is circulated over the social media, it undergoes compression–often multiple times. In order to identify fake audio downstream, our models must be robust to this kind of distribution modification. Additionally, bad actors will try to obscure as much as possible that an audio sample has been faked. A common approach to obscure fake audio is to play to audio aloud in a room and re-record the audio before disseminating it. We create two additional datasets from the ASVspoof dataset for an additional challenge.

**Compressed ASVspoof**   To tackle the issue of compression, we write all of the audio samples in the ASVspoof dataset to the lossy MP3 format (from the lossless FLAC format) with a bitrate of 128k. This shrinks each file, on average, to 33.7% of its original size.

**Rerecorded ASVspoof**   To tackle the issue of re-recording, we re-recorded audio samples in the ASVspoof dataset by playing them aloud on a 2021 MacBook Pro in a large, closed stone-walled room while simultaneously recording.

We also refer to FakeAVCeleb as a well-known benchmark in deepfake audio detection. The FakeAVCeleb dataset is a standard in the deepfake audio detection repertoire, but it is now slightly out-of-date, as it was released in 2021 (Khalid et al., 2021). It contains both deepfake audio and video, but we use only the audio component in this study. The audio subset of the FakeAVCeleb dataset contains $9\,712$ real audio samples and $10\,843$ deepfake audio samples. The language of each audio sample is English, but FakeAVCeleb includes balanced classes of male and female speakers as well as speakers who self-identify as African, East Asian, South Asian, Caucasian (American), and Caucasian (European) (Khalid et al., 2021). This makes FakeAVCeleb the most linguistically diverse of the datasets considered in this study.

## C   HYPERPARAMETERS

Both AST and Wav2Vec transformer models use their own specialized encodings. Otherwise, they are finetuned very similarly. Table 3 reports the hyperparameters used when finetuning the Wav2Vec and AST models, as well as the experiments performed with each model and dataset combination.

### C.1   GBDT

Given its recent popularity for deepfake audio classification tasks Bird & Lotfi (2023); Togootogtokh & Klasen (2024), we use the gradient boosting classifier as the baseline in this study. The features for these tests are the hand-crafted, signal processing features described in Section 3. We use a total of 37 features, which include 20 MFCC features, 12 chroma features, spectral bandwidth, spectral roll-off, spectral centroid, ZCR, and RMS. The features for each audio sample are calculated by averaging each feature's values across all frames. For example, the final ZCR is calculated by averaging the ZCR for each frame across the length of the entire audio sample.

To evaluate best possible performance with the GBDT, we do a hyperparameter search across maximum tree depths and maximum number of estimators. We consider maximum tree depth in $\{3, 8, 10, 15, 25\}$ and number of estimators in $\{10, 100, 200, 400, 600\}$. We do our hyperparameter search with the ASVspoof dataset. We conduct tests with $10\,000$ total data samples, 33% of which are held out as a test set. As GBDTs are highly sensitive to unbalanced data classes, we balance classes before conducting experiments with the GBDT.

As shown in Table 2, the classification accuracy is highest with shorter decision trees and more estimators when trained with 3-second audio samples. The performance stabilizes with $400$ or more estimators, so we take $400$ estimators and maximum depth of $8$ as our optimal hyperparameter setting. We find that this configuration also offers best performance for other audio sample lengths. We observe that performance improves with the number of estimators, but we also see diminishing returns after $400$ estimators. We conclude that $400$ estimators and a maximum tree depth of $8$ are the optimal hyperparameters for this training scenario.

We compute feature importances with the following algorithm (Pedregosa et al., 2011):

Table 2: Accuracy of gradient boosting classifier for various maximum tree depths and number of estimators.

|  | | Number of Estimators | | | | | |
|---|---|---|---|---|---|---|---|
|  | | 10 | 100 | 200 | 400 | 600 | 1000 |
| Max Depth | 3 | 0.748 | 0.822 | 0.838 | **0.850** | 0.856 | 0.859 |
|  | 8 | **0.792** | **0.840** | **0.854** | **0.860** | **0.859** | **0.862** |
|  | 10 | 0.787 | 0.836 | 0.845 | **0.848** | 0.848 | 0.848 |
|  | 15 | 0.745 | 0.791 | 0.814 | **0.814** | 0.814 | 0.814 |
|  | 25 | 0.725 | 0.732 | 0.736 | **0.736** | 0.736 | 0.736 |

---

**Algorithm 1** Permutation Importance Algorithm

---

**Input:** fitted model $m$, dataset $D$, metric $a$, repeats $R$

$\quad s \leftarrow a(m, D)$

$\quad$ **for** each feature $j$ in $D$ **do**

$\quad\quad$ **for** repeat in $1 \dots R$ **do**

$\quad\quad\quad$ Randomly shuffle column $j$ in $D$ to create corrupted $\hat{D}_{r,j}$

$\quad\quad\quad s_{r,j} \leftarrow a(m, \hat{D}_{r,j})$

$\quad\quad$ **end for**

$\quad\quad i_j = s - \frac{1}{R} \sum_{r=1}^{R} s_{r,j}$

$\quad$ **end for**

---

### C.2 TRANSFORMERS

Both AST and Wav2Vec transformer models use their own specialized encodings. Otherwise, they are finetuned very similarly. Table 3 reports the hyperparameters used when finetuning the Wav2Vec and AST models, as well as the experiments performed with each model and dataset combination.

## D MODEL PERFORMANCE ON ASVSPOOF 5 AND FAKEAVCELEB

We evaluate the performance of the gradient boosting classifier on compressed audio samples with the optimal hyperparameters of 400 estimators and maximum tree depth of 8. Figure 6 shows that performance is significantly worsened by both compression and rerecording. The difference in performance is particularly stark on short audio samples, where the gradient boosting classifier excels with the original, unmodified ASVspoof data.

Here, we demonstrate the superiority of the transformer-based methods we consider on the FakeAVCeleb dataset.

Table 3: Hyperparameters used in the finetuning of Wav2Vec and AST transformers.

| Hyperparameter | Value |
|---|---|
| Batch Size | 32 |
| Learning Rate | $3 \times 10^{-5}$ |
| Training Steps | 40 |
| Train-Test Split | 0.33 |
| Weight Decay | 0.0 |
| Warm Up Ratio | 0.1 |
| Optimizer | *AdamW* Loshchilov & Hutter (2019) |

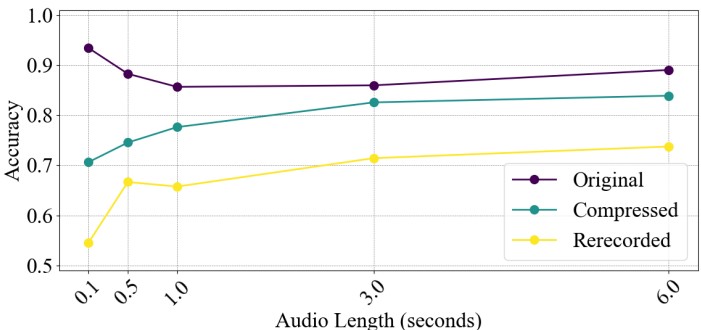

Figure 6: Accuracy over various audio lengths for the original, compressed, and rerecorded audio data for booster graphs with $10\,000$ datapoints, max depth of $8$, and $400$ estimators.

Table 4: Accuracy and ROC AUC comparison of models on FakeAVCeleb.

| Model | Accuracy | ROC AUC |
|---|---|---|
| EfficientNet-B0 (Tan & Le, 2019) | 0.500 | - |
| MesoInception-4(Afchar et al., 2018) | 0.540 | - |
| VGG16 (Simonyan & Zisserman, 2014) | 0.671 | - |
| Xception (Rossler et al., 2019) | 0.763[1] | 0.853 |
| AD DFD | - | 0.881 |
| LipForensics (Haliassos et al., 2020) | - | 0.911 |
| FTCN (Qian et al., 2021) | - | 0.931 |
| AVAD (Gu et al., 2023) | - | 0.945 |
| RealForensics (Zhao et al., 2022) | - | 0.971 |
| FACTOR (Dzanic et al., 2023) | - | 0.974 |
| AST (ours) | 0.979 | 0.985 |
| Wav2Vec (ours) | **0.991** | **0.990** |

## D.1 DATA AUGMENTATION

Though the ASVspoof 5 dataset includes compressed and rerecorded audio samples, they are not marked or isolated within the dataset. To evaluate the impact on performance that these augmentations have, we create three distinct datasets: one with the original ASVspoof 5 data, one with all data compressed, and one with all data rerecorded. We evaluate the effect that these augmentations have on model performance.

---

[1]Unimodal (audio-only) result.

Table 5: Comparing the precision, accuracy, recall, and F1 of the models GBDT, AST, and Wav2Vec for 6.0-second samples of original, compressed, and rerecorded ASVspoof data.

| augmentation | model | precision | accuracy | recall | F1 |
|---|---|---|---|---|---|
| original | GBDT | 0.903 | 0.896 | 0.891 | 0.894 |
| | AST | 0.981 | 0.992 | 0.987 | 0.984 |
| | Wav2Vec | 0.993 | 0.998 | 1.000 | **0.997** |
| compressed | GBDT | 0.853 | 0.841 | 0.837 | 0.841 |
| | AST | 0.994 | 0.994 | 0.982 | 0.988 |
| | Wav2Vec | 0.995 | 0.998 | 1.000 | **0.997** |
| rerecorded | GBDT | 0.589 | 0.589 | 0.589 | 0.589 |
| | AST | 0.968 | 0.991 | 0.994 | 0.981 |
| | Wav2Vec | 0.998 | 0.995 | 0.996 | **0.997** |

# E EXPLAINABILITY

## E.1 GBDT FEATURE IMPORTANCES

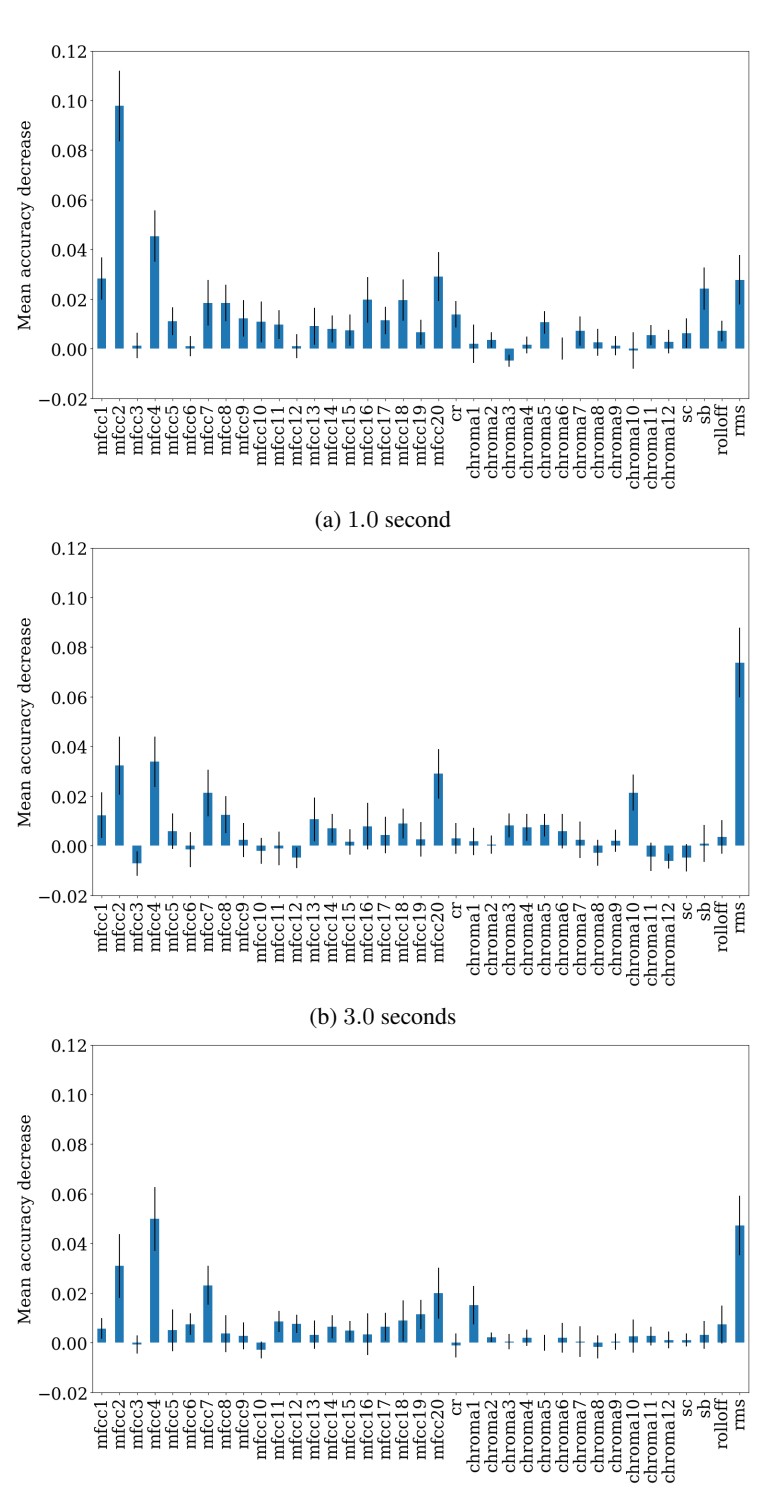

Figure 7: GBDT feature importances as measured by mean accuracy decrease with standard deviations for the 1.0, 3.0, and 6.0-second classifiers.

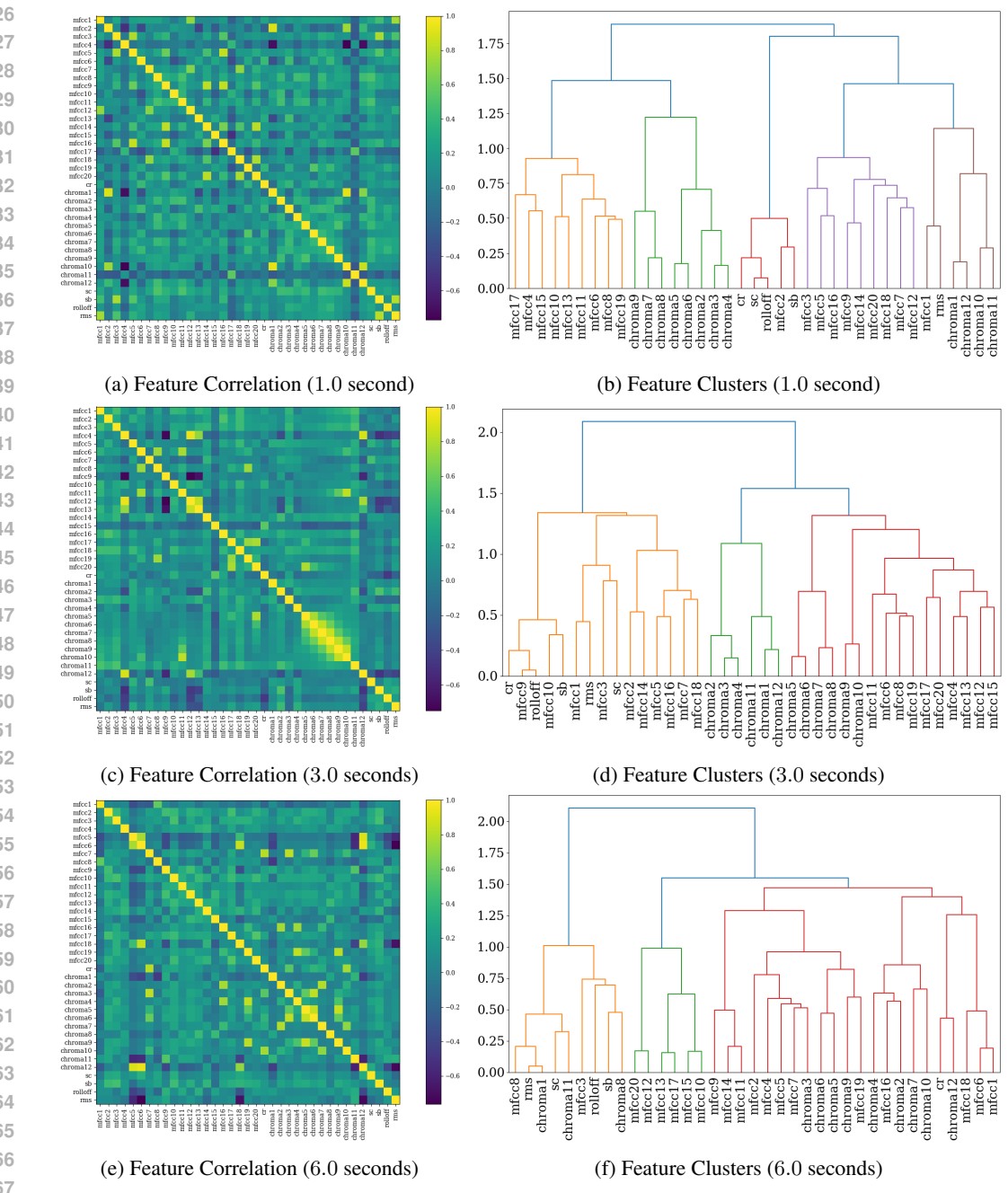

(a) Feature Correlation (1.0 second)

(b) Feature Clusters (1.0 second)

(c) Feature Correlation (3.0 seconds)

(d) Feature Clusters (3.0 seconds)

(e) Feature Correlation (6.0 seconds)

(f) Feature Clusters (6.0 seconds)

Figure 8: GBDT feature correlations and clusters for the 1.0, 3.0, and 6.0-second classifiers. In these figures, sc refers to the spectral centroid, sb refers to the spectral bandwidth, cr refers to the ZCR, mfcc$i$ refers to the $i$-th MFCC feature, and chroma$i$ refers to the $i$-th chroma feature.

## E.2 LAYER-WISE ATTENTION

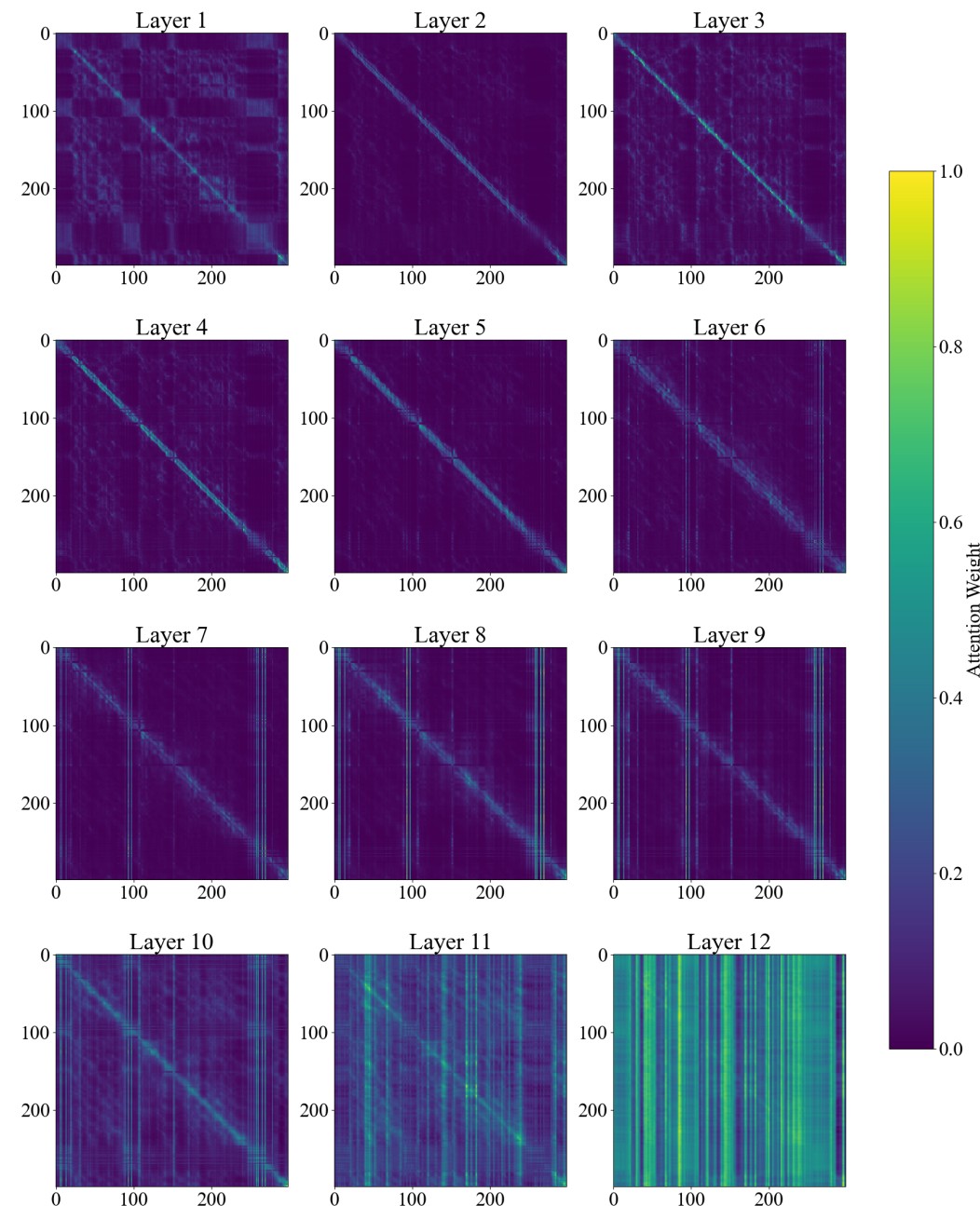

Figure 9: Normalized attention visualized for a bonafide 6.0-second sample where axes represent input token ID.

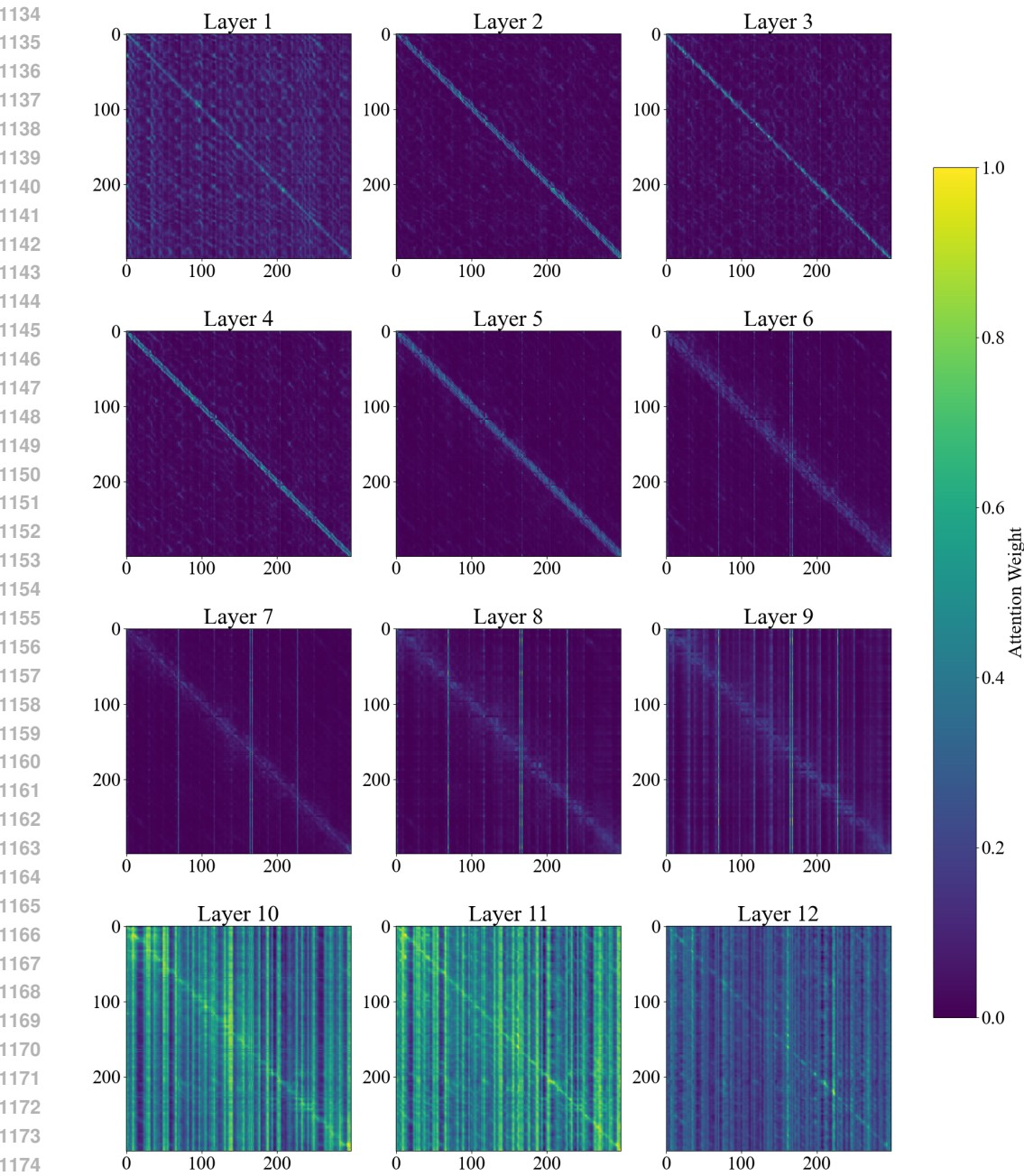

Figure 10: Normalized attention visualized for a spoof 6.0-second sample where axes represent input token ID.

### E.3 Attack Classification

Another potentially useful piece of information is what attack likely generated a given deepfake audio segment. Though this type of "explainability" does not give insight into a model's decision-making, it may provide a clue to the origin of a deepfake audio sample–something likely very important to a journalist or member of law enforcement. We therefore evaluate the abilities of the GBDT and transformer methods to identity which attack, if any, was used to generate an audio sample. Using 6.0-second audio samples, the GBDT achieved an overall accuracy of 51.9% with average precision of 53.0% and average recall of 52.1% over all 18 classes, with relatively stable performance across the attack types. The Wav2Vec model achieved overall evaluation accuracy of 91.8% with average precision of 91.7% and average recall of 91.8% across the 18 classes. The AST model performed similarly, achieving overall evaluation accuracy of 91.1%, average precision of 91.1%, as well as average recall of 91.1%. Though the models are quite successful at this task, previous exposure to deepfake audio generated with each attack is essential. It is not clear how well a model could detect an out-of-distribution sample.

## F    Limitations

While this study introduces a novel benchmark for evaluating the generalization capabilities of state-of-the-art transformer-based models, there are several limitations that need to be addressed to advance the field of audio deepfake detection. First, the reliance on only two datasets, ASVspoof 5 and FakeAVCeleb, limits the scope of our evaluation. These datasets, while diverse, may not encompass the full range of manipulations, recording conditions, and spoofing techniques found in real-world scenarios. This constraint may lead to an overestimation of model robustness and underrepresentation of other spoofing methods. Expanding the benchmark to include more diverse datasets with a wider variety of deepfake techniques would provide a more comprehensive evaluation of model performance and generalization.

Second, the explainability methods employed, such as attention roll-out and occlusion, are still limited in their ability to provide intuitive and human-understandable explanations, particularly for non-technical audiences. While these methods help visualize and highlight the regions of the input that influence the model's decisions, they do not yet offer a complete picture of why a model might fail on specific instances or how it generalizes across domains. Moreover, the explainability results may be sensitive to changes in model architecture and hyperparameters, which could lead to inconsistencies in interpretation.

Lastly, while we explore the generalization capabilities of the proposed models using cross-dataset evaluation, the impact of various environmental factors such as background noise, speaker variability, and language differences were not explicitly analyzed. This could affect the real-world applicability of the models, especially in diverse and dynamic environments. Further research should focus on robustness testing under varying conditions and on creating synthetic data to simulate potential challenges encountered during real-world deployment.

