# OpenReview forum: "Toward Robust Real-World Audio Deepfake Detection: Closing the Explainability Gap"
_ICLR.cc/2025/Conference — Submitted to ICLR 2025_

### Official Review · Reviewer_xF2E · 2024-10-26

**Soundness:** 1
**Presentation:** 3
**Contribution:** 1
**Rating:** 1
**Confidence:** 5

**Summary:**

The paper focuses on the limitations of explainability in current audio deepfake detection methods and provides three-fold contributions to the subject. Firstly, the paper proposes a conceptual explainability framework emphasizing sample-specific, time-specific, and feature-specific explanations interpretable by humans. Secondly, the paper provides empirical evaluations of novel explainability methods for transformer-based detection models, using occlusion and attention visualization techniques. The occlusion method masks out sections of a mel-spectrogram to reveal which parts are most important for final classification. The attention visualization technique and roll-out method are utilized to get a distribution of attention across all layers and can be used to point out parts of the input sequence most relevant for the final classification. Finally, the paper also introduces a novel framework to evaluate the generalizability of deepfake classifiers, where models are trained on one dataset but evaluated on another dataset.

**Strengths:**

1. The paper is very well written, the concepts are easily understood and the limitations of the work are discussed as well.
2. The subject of explainability in audio deepfake detection is a significant and timely problem, especially as deepfake generation technology is readily accessible to the public and has a high capacity of being misused. Research into this subject is very important due to the impact on society this technology can have.
3. The paper makes a difference between interpretability and explainability and proposes that explainable methods should provide interpretable explanations that are sample-specific, time-specific, and feature-specific. This robust definition of explainability would ensure greater applicability of these methods to interpret black-box models.

**Weaknesses:**

1. Both the proposed methods for audio explainability, occlusion and attention visualization, are already existing methods that are very common in literature, especially for vision and language tasks. This is also recognized by the paper, which mentioned that their contribution is porting the methods to the audio task. But to do that, the paper didn't make any modality-specific changes to the methods or edit the methods in any way. For the occlusion method, it's well-known that mel-spectrograms can be treated as images and the paper directly used them on the traditional method. For the attention-visualization method, this is modality agnostic and is based on transformer architecture which takes tokens as input. The roll-out attention method was introduced for natural language tokens and the paper claims to adapt the method for audio tokens, but it's unclear what kind of modifications they did except just replacing the tokens. So the novelty introduced by the paper in these methods is questionable.
2. The paper provided the results of their methods in Figure 4 and Figure 5. First of all, the paper understands that the result in Figure 4 doesn't help explain the model's decision-making (line 420) and instead is suggestive of transformer model behavior. Second of all, the result in Figure 5 is also not very helpful in human interpretation. When these methods are applied to a visual or textual domain, a human can more easily interpret the segment and decision-making rationale. This suggests that more work needs to be done in the audio domain to make the results more human-interpretable. These observations are also recognized by the paper in their limitation section. So the usability of these methods is questionable.
3. The paper claims to introduce a novel benchmark to evaluate the generalization capabilities of deepfake audio classifiers, where they train the model in the ASVspoof5 dataset and evaluate it on the FakeAVCeleb dataset. Here the only contribution of the benchmark is training on one dataset and evaluating on another dataset. This mechanism is already well-known and well-practiced to show the generalization capability of a model to out-of-domain datasets. Here, there is no contribution to the dataset, no new evaluation metric is introduced or any other changes are proposed. Thus, it's questionable to consider this benchmark as a contribution. The abstract also claims to "open-source a novel benchmark for real-world generalizability". The question is what is there to "open-source" here, as the datasets are already available.

**Questions:**

1. What is the technical novelty contribution of the paper's usage of existing explainability methods? Were any changes needed to adapt these existing methods for audio classification? Why not introduce modifications that can take advantage of features specific to audio modality?
2. The result of the explainability methods didn't seem helpful for explaining model behavior. Can you provide more insights regarding this? For example, a concrete deepfake audio sample and its corresponding outputs for explainability and your analysis that will help reason about model behavior? Also, it would be helpful if a scenario is provided where the techniques are useful in practice and have a big impact on our understanding of a model's capabilities.
3. Why is the benchmark considered a novel contribution? This kind of evaluation is very common in deep learning models.

---

> ### Author Response · Authors · 2024-11-25
>
> We thank the reviewer for their helpful feedback.
>
> Regarding the novelty of the contribution: The novelty lies in applying interpretability methods from the visual domain to the audio domain, thus addressing the issue of sparsity in natural language audio descriptions that we identified in collaboration with real-world practitioners.
>
> Regarding the usability of the figures for interpretability: We agree with this criticism and are able to produce audio explanations, but it is difficult to present that in the manuscript. With the final publication of the paper, we plan to publish a website with sample audio tracks.
>
> On the use of "open source" with regards to the benchmark: We recognize that the language we used here was imperfect and will correct it in the camera-ready copy.
>
> We answer the reviewer's questions as follows:
> 1. The novelty is in applying and evaluating interpretability methods common in other domains on the audio domain. As in the papers that adapt Shapley and NMF for audio, we report results to better understand the utility of different interpretability methods for audio, a thus far under-studied application area for interpretability.
> 2. We clarify that the figures provided in the manuscript are sample-specific, but we are unable to present audio interpretations in the manuscript. With the final publication of the paper, we plan to publish a website with sample audio tracks.
> 3. The benchmark is considered a novel contribution because, though other works evaluate their methods on a held-out set with different data provenance, this is not standardized. We hope that by presenting this as a benchmark other works that train on ASVspoof5 will also evaluate their generalizability in way that facilitates comparison with our work and any subsequent work on ASVspoof5.

---

### Official Review · Reviewer_Ee6m · 2024-11-02

**Soundness:** 2
**Presentation:** 2
**Contribution:** 1
**Rating:** 3
**Confidence:** 4

**Summary:**

The paper proposes a novel explainability framework for audio transformers in audio deepfake classification. It proposes utilizing image occlusion to detect feature importance and attention roll-out to understand features better. It also open-sources a novel benchmark for detecting audio deepfakes in real-world cases, which consists of training on ASVspoof5 dataset and testing on FakeAVCeleb dataset.

**Strengths:**

The authors have analyzed the information available from Attention Rollouts and Image Occlusion methods. They also noted that some very short frames from audio signal representation are influential to transformers in classifying and these frames typically appear in groups, which, if further explored, may potentially lead to better interpretability of audio deepfake classifications.

**Weaknesses:**

As the authors themselves have pointed out, attention roll-out and image occlusion-based analysis have been in existence for quite some time, but the novelty of the proposed work lies in applying them in spectrograms to aid in the explainability of audio deepfake analysis. However, how these attention roll-out and image occlusion-based analyses are aiding explainability specific to the audio deepfake analysis is not adequately explained, and how their contribution differs from already existing contributions of attention roll-out and image occlusion-based analysis methods in image feature explainability remains unclear.

They have utilized the occlusion method in an attempt to explain how the model is reaching these decisions, but as they themselves pointed out, it was not helpful in explaining the model’s decision-making. They have also used an attention visualization method and stated that they can attribute specific frames that were instrumental to classification. However, using attention visualization to attribute where a transformer model is putting importance is not novel, and their analysis does not show enough contribution specific to explaining the decision process in audio deepfake classification in transformers.

They have proposed a new benchmark, which consisted of training on one existing dataset and testing on another. They have not provided adequate explanations as to how their novel benchmark would be more helpful in audio deepfake classification, and the idea of testing on a new dataset itself is not particularly novel.

**Questions:**

1. How are these attention roll-out and image occlusion-based analyses aiding explainability specific to the audio deepfake analysis, and how does this contribution differ from the contributions of attention roll-out and image occlusion-based analysis methods in image feature explainability?

2. Are the normalized token attention plots taken as an average of multiple audio samples or the entire dataset? How do these token attention plots vary with different samples? Further information is needed on the normalized token attention plots.

3. It is stated that “..we can pinpoint specific frames that were instrumental in the classification and inspect them more closely…. and we observe that influential tokens typically appear in groups.” This is one of the primary focus of the paper, and further analysis should be done to elaborate these findings.

4. What is the significance of training ASVspoof5 and inferring on FakeAVCeleb over existing benchmarks, beyond the test of generalizability? What specific challenges might inferring on FakeAVCeleb bring that training on ASVspoof5 would not cover?

---

> ### Author Response · Authors · 2024-11-25
>
> Thank you sincerely for your feedback and time reviewing our work. We respond to the reviewer's questions as follows:
> 1. We are happy to include further analysis of this in the camera ready, however, we see the application of non-natural language based XAI methods to audio explainability by itself as a significant contribution to the community.
> 2. The normalized token attention plots are sample-specific. Normalized here means that the attention is normalized across the layers rather than across samples. The attention roll-out allows the viewer to identify frames of the audio that are most salient in classification—and play them back if desired.
> 3. The observation that the influential tokens typically appear in groups gives a sense of the length and importance of the temporal dimension of each prediction.
> 4. The method by which FakeAVCeleb deepfake audio is generated differs from that by which the ASVspoof5 deepfake audio is generated. This means that the type and character of artifacts in the FakeAVCeleb deepfake audio differ from all those in the ASVspoof5 deepfake audio, which demands that the model better understand the characteristics of true audio in comparison to any type of fake audio.

---

> > ### Comment · Reviewer_Ee6m · 2024-11-26
> > **Official Comment by Reviewer Ee6m**
> >
> > Thank you for your response. Unfortunately, most of my concerns have not been addressed. So I decided to keep my score.

---

### Official Review · Reviewer_3hX6 · 2024-11-03

**Soundness:** 3
**Presentation:** 3
**Contribution:** 2
**Rating:** 5
**Confidence:** 4

**Summary:**

This paper addresses the limitations of existing audio Deepfake detection models in terms of generalization ability and interpretability in real-world scenarios. It proposes a novel interpretability approach and establishes a new benchmark to assess model generalization performance. The study trains models on the ASVspoof dataset and evaluates them on the FakeAVCeleb dataset, demonstrating the superior performance of Transformer-based audio detection models on unseen data. Additionally, the paper introduces attention visualization and occlusion techniques to enhance model interpretability, aiming to bridge the gap between model performance and interpretability.

**Strengths:**

1. The paper introduces an interpretability framework for the audio domain, incorporating interpretability techniques from visual and natural language processing into audio Deepfake detection, providing a clearer interpretative path for the model's black-box decision-making process.

2. Two interpretability methods (attention visualization and occlusion techniques) are systematically explored to assess the interpretability of Transformer-based audio detection models, with a comparison of each method's strengths and weaknesses.

3. Cross-validation using the ASVspoof and FakeAVCeleb datasets demonstrates the model's generalization capability across varying data distributions, simulating data shift scenarios common in real-world applications.

**Weaknesses:**

1. The author mentions three limitations in the Limitation section, of which the latter two could serve as directions for future work. However, the first limitation needs to be addressed in the current phase of the task: relying solely on the ASVspoof and FakeAVCeleb datasets may not cover the full range of audio deepfake techniques encountered in real-world scenarios, presenting a dataset limitation. A wider variety of datasets and scenarios is needed to demonstrate the robustness and completeness of the current approach.

2. The introduced attention visualization and occlusion techniques are computationally intensive, potentially affecting the efficiency of practical deployment. Further analysis and comparison of computational costs would be beneficial.

3. The paper’s contribution is somewhat limited, as occlusion and attention visualization are commonly used techniques in computer vision and natural language processing. While adapting these methods for audio Deepfake detection is interesting, the lack of specific modifications tailored to the characteristics of audio forgery detection tasks reduces the overall impact of the proposed approach. Simple method transfer limits the originality of the contribution.

**Questions:**

1. It is recommended to incorporate a wider variety of audio forgery datasets to validate the model's generalization capability across different forgery techniques, aligning more closely with the diversity encountered in real-world scenarios.

2. While interpretability is essential, it would be beneficial to analyze how interpretability can contribute to designing better models or evaluating existing ones. Adding such analysis could enhance the paper's contribution by illustrating the practical impact of interpretability on model improvement and assessment.

---

> ### Author Response · Authors · 2024-11-25
>
> We thank you earnestly for your feedback and helpful suggestions.
>
> Regarding dataset limitations: We selected ASVspoof and FakeAVCeleb because they are publicly available and widely used, but we are happy to include evaluation of additional datasets in the camera-ready copy if the reviewer has some suggestions.
>
> Regarding the requisite time and compute: We do have this data and see the reviewer’s point. We are prepared to include an analysis of the time and computational expense required for our methods in the camera-ready version. We note, however, that we propose these methods for an authority which is studying individual deepfake audio samples closely rather than doing explainability at scale.
>
> Regarding the novelty of the work: While we acknowledge the reviewer’s perspective here, we want to highlight that applying these methods to the audio domain is novel and is thus far unreported. We hope that the publication of our work facilitates additional work in this area without the necessity of repeating what we have already done.

---

### Official Review · Reviewer_nNp8 · 2024-11-04

**Soundness:** 2
**Presentation:** 1
**Contribution:** 1
**Rating:** 1
**Confidence:** 4

**Summary:**

The paper analyses the existing explainable methods, such as Occlusion and Attention visualization, for deepfake audio detection tasks. The authors show results using three baseline models such as AST, GBDT, Wav2Vec. The authors evaluate these methods on two existing datasets.

**Strengths:**

- Authors provide an analysis of explainable and interpretable methods for audio deepfake detection methods. The authors compare these methods on two different benchmark datasets.

**Weaknesses:**

- The paper is merely an analysis paper of existing explainable methods. There are no significant novel contributions to this paper. The authors mention in line 052 that the contributions are "Empirical evaluations of novel explainability methods for audio transformers". However, I cannot judge what novelty is in attention visualization for transformers. It has happened a lot in literature.
- When authors compare explainable methods for audio deepfake detection, they must show the results on existing approaches such as Shapley[1].
- The authors should include more explainable mechanisms as baselines, such as [2], [3].
- The authors should show results on multilingual deepfake audio datasets such as MLAAD, DECRO, and WaveFake since AST and Wav2vec are pre-trained on the AudioSet dataset, primarily English. Evaluating a multilingual dataset would help strengthen the analysis.
- Authors show results on the ASVspoof5 dataset. The authors mention that the dataset was released in June 2024 (line 750). However, The dataset is still not available for public use and review. Authors should show the results on the publicly available datasets.
- FakeAVCeleb dataset is primarily an audio-video deepfake dataset, not only for audio deepfake detection. The FakeAVceleb dataset contains 500 real videos, which means there should be 500 real audio. However, authors in line 7716 mention 9712 real audio samples. Why is there a difference in the numbers?
- Even the baseline models such as GBDT, Wav2Vec and AST are general audio architectures, not specifically designed for audio deepfake detection. Authors must show results on models such as ASSIST, RawGAT-ST and some state space models such as RawBMamaba.
- More then 50% of the paper is just explaining trivial and non paper contributions only.



[1] Explaining deep learning models for spoofing and deepfake detection with SHapley Additive exPlanations
[2] Listen to Interpret: Post-hoc Interpretability for Audio Networks with NMF
[3] Focal Modulation Networks for Interpretable Sound Classification

**Questions:**

Please see the weakness listed above.

---

> ### Author Response · Authors · 2024-11-26
>
> We thank the reviewer for all of their helpful feedback and suggestions. We respond to the reviewer's comments as follows:
> 1. The novelty lies in applying interpretability methods from the visual domain to the audio domain, thus addressing the issue of sparsity in natural language audio descriptions that we identified in collaboration with real-world practitioners.
> 2&3. Since our initial submission, we have explored alternative methods for interpreting audio, including Shapley and LIME, which we are prepared to include in the camera-ready version. In the camera ready copy, we will contrast what each interpretability method attributes model decision-making to.
> 4. We recognize that this would strengthen the generalizability benchmark, but we do not expect that the artifacts of deepfake audio will vary meaningfully on the language of the deepfake audio. Does the reviewer have reason to believe that is likely?
> 5. We do show results on the publicly available and widely popular FakeAVCeleb dataset.
> 6. In this study, we operate on 6-second audio samples. The length of the samples in FakeAVCeleb is not uniform, so, when segmented into 6-second samples, there are unbalanced class sizes. Though FakeAVCeleb is an audio-video dataset, it has been used in prior work for its audio only (see Faceforensics++: Learning to detect manipulated facial images by Rossler et al.)
> 7. We appreciate this suggestion and consider it a very interesting avenue to explore. We plan to deliver results on these models as well in the camera-ready copy to better contextualize our results with the general audio architectures. We thank the reviewer for this suggestion.
> 8.  We see a comprehensive review of audio interpretability as part of our contribution, as we are trying to address a long-standing problem that has so far not been comprehensively addressed.

---

> > ### Comment · Reviewer_nNp8 · 2024-11-27
> >
> > Thank you very much for your responses. Since most of my queries remain unanswered, I decided to keep my score.

---

### Meta-Review · Area_Chair_GdGW · 2024-12-17

**Metareview:**

This paper generally provides an open discussion and analysis of explainable and interpretable algorithms for audio deepfake detection. All the reviewers thought of the low-quality technical contributions. I would like to quote some words from the four reviewers as follows:
As Reviewer nNp8 pointed out, this paper is merely an analysis paper of existing explainable methods and there are no significant novel contributions to this paper.
As Reviewer 3hX6 questioned, the contribution of this paper is somewhat limited, as occlusion and attention visualization are commonly used techniques in computer vision and natural language processing.
The Reviewer Ee6m has clear concerns about the contribution, token attention plots, the significance of training, and others.
Reviewer xF2E raised the concerns about the technical novelty contribution of the paper's usage of existing explainability methods.

After rebuttal, all the reviewers did not promote their rating scores. Therefore, based on the comments, rebuttal, and discussion, this paper should be rejected without further consideration in ICLR.

**Additional Comments On Reviewer Discussion:**

All the reviewers voted to reject this paper, even after rebuttal and discussion.

---

### Decision · Program_Chairs · 2025-01-22

Reject